# Reasons Behind (Un)Healthy Eating Among School-Age Children in Southern Peru

**DOI:** 10.3390/foods14030348

**Published:** 2025-01-21

**Authors:** Karina Eduardo, José Carlos Velásquez, Jhony Mayta-Hancco, Juan D. Rios-Mera, Michelle Lozada-Urbano, Erick Saldaña

**Affiliations:** 1Sensory Analysis and Consumers Study Group, Escuela Profesional de Ingeniería Agroindustrial, Universidad Nacional de Moquegua, Prolongación Calle Ancash s/n, Moquegua 18001, Peru; keduardop@unam.edu.pe (K.E.); 2018102039@unam.edu.pe (J.C.V.); jmaytah@unam.edu.pe (J.M.-H.); 2Instituto de Investigación de Ciencia y Tecnología de Alimentos (ICTA), Universidad Nacional de Jaén, Jaén 06800, Peru; juan.rios@unj.edu.pe; 3Universidad Privada Norbert Wiener, Lima 15046, Peru; michelle.lozada@uwiener.edu.pe

**Keywords:** descriptive projective technique, healthy foods, nutritional knowledge, sensory methods, Peruvian children

## Abstract

In recent decades, rates of childhood overweight and obesity have increased worldwide, surpassing those of adults. Understanding the factors influencing children’s food choices is essential to promote healthy eating habits. This study examined why school-aged children make healthy and unhealthy food choices and how parents’ eating habits influence their children’s choices. Children’s ability to rank 25 food pictures according to their healthfulness was assessed using a free sorting task (FST), while parents made a free list of healthy and unhealthy foods, and the cognitive salience index (CSI) was calculated. Children were grouped into two groups based on their responses: one mainly from Moquegua and the other from Puno. In general, children from Moquegua demonstrated a greater ability to identify healthy foods than those from Puno. The CSI provided information on healthy and unhealthy foods in each region. These findings underline the potential of selecting palatable and healthy foods to improve children’s diets in the short term and contribute to the development of healthier products in the food industry. In addition, studies in other regions of Peru are recommended to understand children’s perceptions better.

## 1. Introduction

Obesity is currently one of the biggest public health concerns worldwide. According to WHO [1], obesity has nearly tripled since 1975, with more than 650 million obese. In the last two decades, obesity rates in children have surpassed adults [2,3,4]. It is estimated that by 2025, the number of children with obesity will exceed 70 million [1], and this number will continue to grow by 2030 [5]. A similar trend is observed in Peru, which ranks 42nd in the world with 19.25% obesity in school-age children and adolescents (5–19 years) and 48th with 14.71% obesity in girls in the same age category [6], with 9.6% of children under five years of age being obese [7]. Childhood obesity can lead to chronic diseases in adulthood, such as type 2 diabetes, cardiovascular disease, and musculoskeletal disorders [8,9]. The rise in non-communicable diseases, such as obesity, has led consumers to choose healthier and more sustainable diets [10]. In this context, it is necessary to understand the reasons behind children’s food choices to develop strategies that promote healthier feeding [11,12,13,14].

Food choice in children is a multifactorial phenomenon influenced by intrinsic factors such as sensory, physiological, genetic, and affective characteristics [15], experience, social influences, and extrinsic factors (cognitive and contextual effects) explaining food choices in adulthood [16,17,18,19,20]. In this framework, the following question arises: will children be able to classify healthy and unhealthy foods based on their knowledge? According to the scientific literature, school-age children can distinguish between healthy and unhealthy foods [21,22,23]. Nevertheless, the reasons for the discrimination remain unclear; it may be that the free sorting task and cognitive salience index (CSI) about foods in two cities with cultural differences are determining factors in food choices, and it would be beneficial to conduct more methodological studies to determine what separates healthy foods from unhealthy ones in children’s minds.

The study of consumer food choice can be conducted through various sensory methodologies, both descriptive and affective approaches [24]. These sensory methods should be modified to ensure that children understand the instructions and are not distracted during the execution [25]. A descriptive projective technique recently adapted for the study of children’s food choices is the free sorting task (FST), which involves sorting foods into categories based on their similarities and differences [26,27,28]. Adaptations to the FST included changing the food stimuli to visual stimuli and using predetermined groups rather than FST. The procedure also recreated a game in which drawings and the task characterized the groups, consisting of placing images of food into the appropriate category [12,29]. Based on these authors’ findings, school-age children could understand the sensory technique and sort their food consistently.

We know that no studies have investigated the relationship between children’s food choices and their parents’ eating habits or the cultural aspects of foods in Southern Peru. A study [30] stated that parents’ eating habits offer a “window of opportunity” to promote healthy eating habits. In this context, this study aimed to identify the reasons behind school-aged children’s classification of healthy and unhealthy foods and evaluate the influence of parental eating habits and cultural context on these perceptions to contribute to developing strategies that promote healthy eating habits in this population.

## 2. Materials and Methods

The parents signed a consent form before conducting these experiments, permitting their children to participate in this research. Detailed information about this study, including procedures and potential risks, was provided before obtaining consent.

This study has been registered by the Dirección de Innovación y Transferencia Tecnológica of the Universidad Nacional de Moquegua (INFORME Nº 0052-2022-UPII-DITT-VPI/UNAM) and approved by the ethics committee of the Universidad Nacional de Moquegua (Carta 001-2024 PCEIC/UNAM).

### 2.1. Food Selection

Based on the diet of the Moquegua and Juliaca populations, 25 model foods were selected using the “free word association” task, in which 30 participants provided information on the foods they consumed in the last two months. Afterward, two researchers classified the foods into “healthy”, “fairly healthy”, and “unhealthy” based on their nutritional composition [31] and the NOVA classification [32,33,34]. Table 1 shows the distribution of the 25 most consumed foods in Moquegua and Juliaca, classified into three unbalanced groups. Representative images of the food were selected and printed on laminated paper 7.9 cm long and 5.4 cm high.

### 2.2. Formal Procedure

This study involved observing children and their parents to analyze the children’s behavior based on the parents’ behavior.

#### 2.2.1. Children

This study recruited 100 children (41% girls and 59% boys) aged 6 to 12 from Moquegua and Juliaca in the southern Peruvian states of Moquegua and Puno, respectively. The cities were chosen for their cultural and geographical diversity in the south region of Peru, enabling an intriguing comparison of children’s eating habits and perceptions.

#### 2.2.2. Children’s Procedure

The structured FST was carried out by ten children, as per the methods used in previous studies [12,29], with some modifications. Instead of using an A3 sheet, the task involved the use of four cardboard boxes labeled with different symbols: an angel (representing good), a devil (representing bad), a happy face (representing tasty), and a sad face (representing not tasty).

These symbols were combined to form four categories: an angel and a happy face (representing healthy and tasty); an angel and a sad face (representing healthy but not tasty); a devil and a happy face (representing unhealthy but tasty); and a devil and a sad face (representing unhealthy and unappealing).

A preliminary pilot session was conducted with 30 children aged 6–8 to confirm their understanding of the symbols and ensure their effectiveness. During this session, children were asked to describe the meaning of each symbol and perform a free sorting task. The process ensured that the symbols were age-appropriate and culturally relevant, allowing for accurate sorting during the main study.

Before the test, the interviewers used cartoon images to explain and familiarize them with this process. The explanation involved classifying images into four boxes labeled with specific symbols. Once prepared, the children were given food images to place in one of the four boxes (see Figure 1). The researchers closely monitored the children to ensure they followed the instructions. After completing the test, each child received a small gift. Data were collected manually after each child’s structured FST.

#### 2.2.3. Parents’ Procedure

The “free listing” technique was used, which involved participants listing items related to a topic without restrictions or external influence [35]. A total of 100 parents, 60 from Moquegua and 40 from Juliaca (corresponding to their children of Moquegua and Juliaca), were interviewed individually as part of this study on children’s food preferences.

They were given the following instruction: “Good morning; please mention four foods that you consider healthy and four foods that you consider unhealthy”. The parents were then asked to list their choices verbally while an interviewer recorded each food on paper in the order mentioned. Parents completed the activity once the children had finished structured FST.

#### 2.2.4. Socio-Demographic Characteristics of Parents

At the end of the free listing task, parents in Moquegua and Juliaca completed socio-demographic data through individual interviews with closed-ended questions on age, educational level, average monthly income, and number of children.

### 2.3. Data Analysis

The multivariate Distatis technique [36] was applied to an organized matrix with samples in the rows and all participants in the columns to classify children’s foods. Using the resulting coordinates of the participants, Euclidean distances were calculated, and a hierarchical cluster analysis was performed to identify participant clusters with the Ward agglomeration criterion. Then, the Distatis technique was re-applied to each cluster to represent the foods in a factorial plane, achieving more homogeneous representations and reducing interindividual differences among the participating children.

The results of the “free listing” of parents were analyzed in two steps. The first step involved the raw data classification. Firstly, two researchers unified foods, such as simplifying brand names like “Coca-Cola” or “Inka-Cola” to “soda”. Secondly, identical foods known by various names were consolidated. For instance, different types of cakes, such as chocolate cake, cupcakes, and cake, were collectively labeled as “cakes”.

After categorizing the different foods mentioned by parents, the total number of foods, the frequency of mentions, and the average position of each food were calculated using Equation (1).(1)$$Ap_{j}=\frac{\sum_{i=1}^{N}R_{ij}}{F_{j}}$$
where $Ap_{j}$ = average position of food *j*; $R_{ij}$ = rank, given by respondent i to food *j*, and $F_{j}$ = number of respondents who mentioned the food *j*. Finally, the CSI was calculated using Equation (2).(2)$$CSI_{j}=\frac{F_{j}}{N\times Ap_{j}}$$
where $F_{j}$ = number of respondents who mentioned the food *j*; N = a total number of respondents, and $Ap_{j}$ = average position of a food *j*. The $CSI_{j}$ was plotted for each food item using a barplot by cluster.

All data analyses were conducted in RStudio software, version 2021.06.1, with DistatisR and ExPosition [37] packages.

## 3. Results and Discussion

After receiving explanations from the interviewers, all the children could perform the structured FST. During the task, it was observed that the children were having fun, as it resembled a game. The children completed the task in between 15 and 20 min. In agreement with the study by Varela and Salvador [29], it is highlighted that from age 5, children can understand and execute the structured FST successfully, and the child’s age influences the time required.

### 3.1. Children Clusters

The children participating in this study were grouped based on Euclidean distances calculated from the food image selections (Figure 2). Through this dendrographic analysis, two differentiated clusters were identified. “Cluster 1”, identified in blue and located on the right, comprised 55 children, of which 47 were from Moquegua, representing 85.4% of the total participants in this cluster. This shows that cluster 1 was mainly composed of children from Moquegua. “Cluster 2”, identified in orange and located on the left, grouped 45 participants. In this case, 32 children were from Juliaca, representing 71%, while 29% were from Moquegua. This indicates that cluster 2 was mainly composed of children from Juliaca, highlighting the geographic and cultural differences between the clusters. Each cluster reflects a specific group of participants characterized by similarities in food preferences and selection patterns. These results reinforce that food preferences and geographic origin were essential in forming the observed clusters.

### 3.2. Classification of Foods

The Distatis technique was used to obtain the two-dimensional representation of “Cluster 1”, which shows 50% of the variability present in the data (Figure 3). This projection reflects 35% and 15% of the variance in the first two dimensions. Thus, the data are well represented in the factorial plane.

Figure 3 shows three groups; on the right side, there are the healthy foods, while on the left, there are the less healthy foods. Foods of intermediate health are located in the center of the plot. The small circles and tight groupings reflect a clear consensus among the children in classifying certain foods as healthy. In contrast, larger circles, as in the case of bread, indicate discrepancies in their classification.

The first group, positioned in the positive part of the first dimension, comprises prickly pear, fish, papaya, lettuce, banana, rice, and milk. According to the NOVA classification [32], these foods would be in group 1, called unprocessed or minimally processed foods. A classification of healthy foods according to their nutritional quality is evident, highlighting the inclusion of fruits and vegetables. This classification is supported by research analyzing dietary patterns and their impact on health, as Olsson et al. [38] note that fruits and vegetables are characterized by their low energy density and high content of micronutrients, fiber, and polyphenols, which may contribute to a possible protective effect against chronic non-communicable diseases.

In Peru, rice is a staple food, with an annual per capita consumption of 74.0 kg and a daily consumption of 0.20 kg per person [39]. Figure 3 shows that children sorted rice as a healthy food, possibly because it was part of their daily diet in Moquegua and Juliaca, which could be influenced by cultural factors present in the household. This finding is consistent with the research of Haines et al. [40], who argue that parental beliefs and cultural influences impact children’s eating habits.

The second group is located in the negative part of the first dimension and is associated with unhealthy or low nutritional quality foods. These include soda, hamburger, chocolate, salchipapa (Peruvian dish consists of french fries and fried sausages), pollo a la brasa (Peruvian baked chicken), cakes, ice cream, candies, cookies, and potato chips. According to Monteiro et al. [32], these foods fall under the fourth group of the NOVA classification, comprising ultra-processed foods. These low-nutrient, high-calorie, high-fat food products may contribute to the development of chronic non-communicable diseases [41,42,43].

The last group, located in the negative part of the second dimension associated with intermediate nutritional quality, consisted of chicken, potato, noodles, bread, quinoa, and yogurt. According to the NOVA classification, they would be in group 1: unprocessed or minimally processed foods, and group 3: processed foods. It is observed that the children classified quinoa within this group possibly due to the low level of consumption, lack of knowledge of its nutritional properties, lack of exposure to this food at home, or unpleasant experiences due to its bitter taste, as indicated in the study by Lima et al. [44], who found that taste, food characteristics, and lack of knowledge about healthy options constitute barriers to an adequate diet.

Children classified foods such as fruit, rice, and milk as “healthy”, particularly when these were consumed regularly at home or school. Conversely, foods considered more palatable, such as sweets, chocolates, and fast food, were classified as “unhealthy but tasty”. This reflects the salient role that the sensory part plays in food choices, prioritizing taste over nutritional content.

Based on the results shown in Figure 3, children in Moquegua appear to understand the healthiness of foods, as demonstrated by their ability to sort them accordingly. These results are similar to those reported by Moura and Aschemann-Witzel [30], indicating that children can distinguish between healthy and unhealthy foods. This outcome may also reflect the influence of parental food patterns on their children’s habits. Studies by Mahmood et al. [45] and Tenjin et al. [46] suggest that children model their food choices based on observed behaviors at home.

Figure 4 shows that Cluster 2, composed mainly of children from Juliaca, presents small circles, such as pollo a la brasa, cakes, and cookies, indicating a greater consensus among children in classifying these foods. In contrast, the larger circles, such as those corresponding to potato chips, sweets, and quinoa, reflect greater variability in individual children’s perceptions. This cluster did not present defined sorting according to the healthiness or nutritional quality of the foods. This situation could be due to the children’s lack of knowledge about the classification of healthy foods or because they did not accurately understand the dynamics of the procedure. Educational interventions in schools can provide children with opportunities to learn about nutrition, improving their ability to identify healthy and unhealthy foods [47].

According to the Instituto Peruano de Economía, Moquegua in Peru has the highest educational attainment, with a score of 8.6, while Puno, which includes Juliaca, has a lower score of 4.5 [48]. This disparity in educational levels may influence the effectiveness of nutritional education, potentially impacting children’s dietary habits and perceptions of food. Children from Moquegua demonstrated a better understanding of the healthy food ranking task than those from Juliaca.

### 3.3. Free Listing of Parents

Parents in Cluster 1, mainly comprising Moquegua, mentioned a wide range of healthy foods, identifying 56 healthy foods. For this purpose, foods mentioned more than three times were considered, comprising 40%. As for unhealthy foods, parents mentioned 32 foods, and those mentioned more than three times were considered. On the other hand, in Cluster 2, mainly comprising Juliaca, parents mentioned 19 healthy foods and 18 unhealthy foods.

A CSI was determined based on the frequency and perceived importance of each food in parents’ minds. Appendix A shows the CSIs calculated for each food item, including their frequencies, average positions, and CSI. For example, in the case of fish, the average position (*Apj*) was calculated using Equation (1). The Sum of Ranks, which corresponds to the sum of the positions assigned by the participants to fish, was equal to 52. This value was divided by the frequency, representing the number of participants who mentioned the fish (23), obtaining an average position value of 2.26. Subsequently, CSI was determined using Equation (2). For this purpose, the frequency (23) was divided by the product of the total mentions (considered as 40% of the total, equal to 22) and the Average Position value (2.26), resulting in a CSI of 0.46. This score is the highest among the foods analyzed, emphasizing its CSI for participants compared to other foods like potatoes and wheat.

In addition, foods such as stews, quinoa, fruits, vegetables, and milk have relatively high CSI ratings. Similar results were reported by Muñoz et al. [49], where fruits, vegetables, legumes, and fish were the most mentioned and trusted foods by pregnant and breastfeeding women in Spain. Likewise, in a study carried out by Barone et al. [50] on sustainable foods, vegetables presented an intermediate CSI similar to this paper’s. In addition, fruits, such as bananas and oranges, were the most commonly mentioned by parents, which coincides with the results reported by Hough and Ferraris [35]. They are considered important in parents’ diets, and consequently, they are significant in children’s diets as well.

Figure 5a depicts various foods and their CSIs. In Figure 5b, certain food items such as “industrialized sauce”, “industrialized juices”, and “chocolates” have been found to have the highest CSI ratings as compared to “oil”, “sausage”, and “noodles”. These foods generally contain high sugar, saturated fat, and sodium levels. They are classified as ultra-processed, according to Monteiro et al. [33]. Consumption of such processed foods has been linked to an increased risk of chronic diseases like obesity, type 2 diabetes, and cardiovascular disease [8,41,43,51].

An association between parents’ food choices and their children’s food classification has been observed in Moquegua. Both parents and children can distinguish between healthy and unhealthy foods. For instance, both groups categorized fish, vegetables, and fruits as healthy foods and chocolates and sweets as unhealthy. This ability highlights the importance of food literacy for parents, which is crucial in promoting healthy eating habits among children from an early age.

In the second cluster, mainly comprising Juliaca parents, the category of healthy foods (Figure 6a) showed “liver” as the food with the highest CSI. It is followed by lettuce, olluco (*Ullucus tuberosus*), and beans. According to parents, these foods are essential in their diet. On the other hand, foods such as lentils, avocados, and meat have lower CSI, indicating lower perceived importance. In unhealthy foods, candies, chocolate, bread, and cookies have the highest cognitive salience index, while cakes, industrialized juice, and hamburgers have a lower CSI. Similar results were reported by Muñoz et al. [49] when pregnant and breastfeeding women considered this type of food unreliable.

Parents in Juliaca could differentiate between healthy and unhealthy foods, indicating that liver, lettuce, and olluco (*Ullucus tuberosus*) are healthy, while sweets, chocolates, and grilled chicken are unhealthy foods. However, their children could not adequately classify healthy and unhealthy foods, possibly due to a lack of full understanding of concepts related to health and nutrition. Nevertheless, it is noted that some fruits and vegetables, such as papaya and lettuce, are very close in their rankings.

Among the healthy foods mentioned in both states, some common ones include quinoa, liver, wheat, lentils, meat, and potatoes. However, the perceived importance of these foods differs between them. Each state also has its unique foods; for example, in Moquegua, fish and fruit are commonly consumed, while in Juliaca, olluco *(Ullucus tuberosus*), oca (*Oxalis tuberosa*), chuño (freeze-dried potato), isaño (*Tropaeolum tuberosum*), and trucha (*Oncorhynchus mykiss*) are popular. These differences in food choices reflect each region’s distinctive cultural and geographical contexts. Moquegua is located in Southern Peru at 1410 m above sea level (m.a.s.l.), situated 89 km from the coast, while Juliaca is positioned in the Peruvian highlands at 3830 m.a.s.l., situated 400 km from the coast [52]. Both states have a significant presence of unhealthy processed foods high in sugar, fat, and sodium. These foods include candies, cookies, chocolate, and soda. Interestingly, rice was also considered unhealthy, possibly due to its carbohydrate and low fiber content [31]. However, in Moquegua, children considered rice to be a moderately healthy food. This difference in perception could be influenced by the frequent presence of rice in their meals, which could shape the way they perceive its healthiness. As Domínguez et al. [53] point out, familiarity with the context may be crucial in revealing the effect on food choices, suggesting that the regular consumption of rice in this region influences children’s perception of its nutritional quality.

Parents significantly influence their children’s diets [54]. In both Moquegua and Juliaca, children classified foods according to what they typically consumed at home. For instance, parents in Moquegua often mentioned fish, vegetables, and fruits as healthy options, which aligned with the children’s classifications. However, adopting healthy eating is hindered by various time constraints, educational backgrounds, modern lifestyles, and socioeconomic inequalities [55].

### 3.4. Socio-Demographic Characteristics of Parents

Table 2 shows the socio-demographic data of the participating parents, highlighting differences between Moquegua and Juliaca in terms of education, income, and household size, all factors that may influence the dietary choices of parents and children.

The educational levels of parents in Moquegua are higher than those in Juliaca, with 64% of Moquegua parents having completed university studies, compared to only 22% in Juliaca. This difference in education may directly influence household eating habits. As Gonzales et al. [56] emphasize, a higher level of education facilitates a better understanding of food quality, allowing for differentiation between healthy, industrialized, and ultra-processed products. This enables better-informed parents to prioritize nutritious foods and encourage balanced eating habits for their children. This, in turn, promotes healthier practices that benefit the entire family. This influence was reflected in the results of the Free listing, where parents in Moquegua mentioned 56 healthy foods and 23 unhealthy foods. In Juliaca, the list was less diverse, with only 19 healthy foods and 18 unhealthy foods mentioned. This could be attributed to a more limited understanding of food health concepts among parents in Juliaca, influenced by educational and economic factors.

In Moquegua, where parents had higher levels of education, children showed a better ability to distinguish between healthy and unhealthy foods. As Garg [57] highlights, parental nutritional literacy plays an essential role in the acquisition of healthy eating habits in children. For this, it is necessary to promote the consumption of familiar foods as a strategy to improve healthy eating patterns [58].

Parents in Moquegua reported higher average incomes than those in Juliaca, which may impact their access to healthy foods at home. Families with higher incomes tend to have a greater ability to purchase fresh and nutritious items, such as fruits and fish, foods that children in Moquegua consider healthy. Similar results have been reported by Wolfson et al. [59], who highlight the relationship between higher incomes and a better ability to purchase fresh and nutritious foods.

In contrast, Juliaca’s lower income may explain the reliance on affordable traditional and processed foods. As noted by Puddephatt et al. [60], income level determines food choices, as families with limited resources prioritize cost over quality, which negatively affects their health and well-being.

This economic disparity is also reflected in food prices, which play a crucial role in decision-making about food. In Moquegua, healthy foods such as fish, legumes, and quinoa have average prices of $2.21/kg, $2.35/kg, and $3.89/kg, respectively, making them more accessible to families with higher incomes. Conversely, in Juliaca, cheaper traditional foods predominate, such as liver ($2.81/kg), oca ($0.8/kg), and olluco ($0.93/kg). Fresh foods, such as fruits ($1/kg), were less frequently mentioned due to low consumption. These prices, collected from local markets, are detailed in Appendix A, offering a representative perspective on the economic and food dynamics in both regions. Additionally, unhealthy foods such as chocolates and candies, which are similarly priced in both regions ($0.26 per piece and $0.08 per piece, respectively), were widely reported during free-listing exercises. This reliance on cheaper, less nutritious options may limit children’s exposure to healthier foods, thereby reducing their ability to recognize and classify them appropriately. These findings highlight the interaction between income, food prices, and dietary habits, particularly in economically disadvantaged households.

According to national data, Moquegua state has monetary poverty levels between 12–16%, while Puno state is among the five departments in Peru with poverty levels above 40% [61]. These economic differences influence the dietary choices of families. In the context of lower income, families tend to opt for more affordable foods with lower nutritional quality, affecting children’s diet and nutritional development.

The larger number of family members in Juliaca than in Moquegua may influence household food choices. Larger families often prioritize more accessible meals, which can limit the variety of foods available to children. In contrast, families with fewer members, such as those in Moquegua, may have more resources to invest in diverse meals, enhancing children’s understanding of healthy eating.

In the same way, creating healthy food environments, such as serving fruits and vegetables in small proportions, help the preference for these, as confirmed by Wang et al. [62] and Vari et al. [63]. Nutrition literacy and active child participation in food preparation [64,65] are key factors that can strengthen children’s cognitive abilities to sort foods and make healthier choices shortly.

External factors, such as advertising of processed foods [66] and strategic packaging design [67], influence food choices, especially among children from lower-income households. As noted by Caton et al. [68], repeated exposure to ultra-processed foods reinforces their appeal, underscoring the urgency of implementing policies that limit their promotion and encourage healthier, wellness-oriented food environments. These external factors add to economic and social influences, highlighting the need for a comprehensive approach that addresses all these aspects to improve children’s diets. In regions such as Juliaca, it is recommended that educational strategies increase exposure to healthy foods and reinforce nutritional knowledge. In regions such as Moquegua, programs can focus on maintaining and expanding healthy eating habits. This study examined two regions of Peru, which may not fully capture the diversity of dietary habits and cultural contexts throughout the country. For this reason, it is recommended that more regions be included in future research.

## 4. Conclusions

The FST effectively demonstrated how parental education and habits influence children’s perceptions of healthy foods. Children’s ability to classify food is crucial for establishing healthy eating habits early in life. Notably, Cluster 1, primarily from Moquegua, excelled at categorizing food images by healthiness. This success reflects the positive impact of higher parental education and nutritional literacy, while Cluster 2 struggled with this task. Children’s food ratings correlated with their parents’ preferences, as they identified fish, fruits, and vegetables as healthy options. This highlights the role of cultural and geographic factors in shaping their perceptions.

Culturally relevant interventions are essential to address these disparities. Hands-on activities, such as food sorting games and cooking workshops, can help children consistently identify healthy choices that align with regional preferences, thereby improving their habits and those of their families. Additionally, workshops for parents should aim to enhance nutritional literacy and encourage incorporating traditional healthy foods into family diets tailored to each region’s cultural and socioeconomic context. Finally, policies that promote healthy food environments, such as reducing advertising for ultra-processed foods targeting children and encouraging exposure to healthier options, are vital for fostering sustainable eating habits across generations.

## Figures and Tables

**Figure 1 foods-14-00348-f001:**
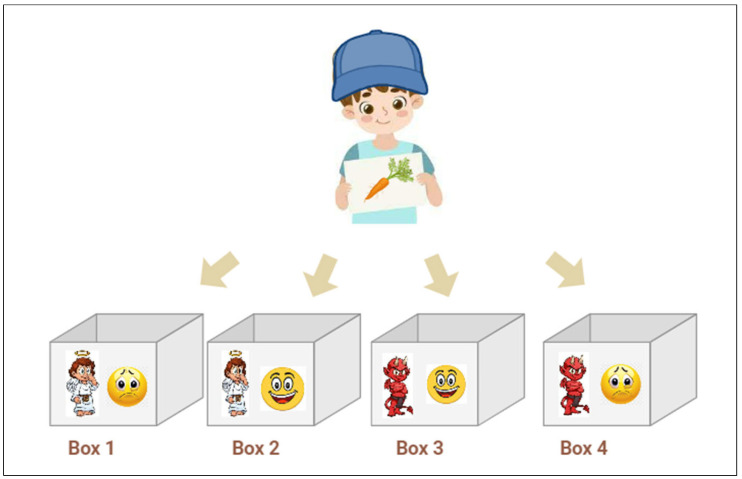
Adaptation of the structured FST using food images and predefined categories, based on Varela and Salvador [29].

**Figure 2 foods-14-00348-f002:**
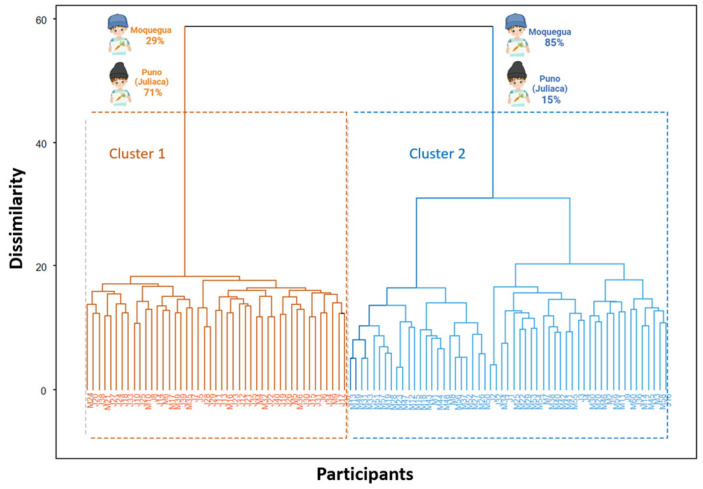
Dendrogram of participating children based on Euclidean distances (own elaboration).

**Figure 3 foods-14-00348-f003:**
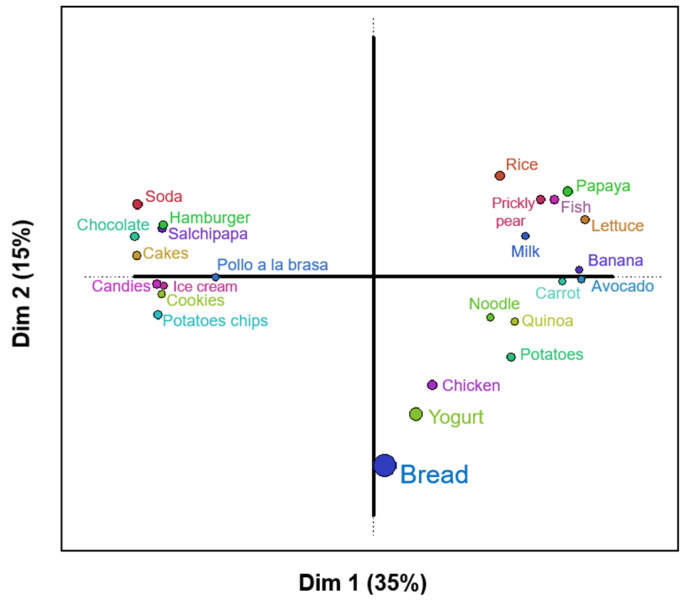
Two-dimensional representation of the foods grouped by the children in Cluster 1 (own elaboration).

**Figure 4 foods-14-00348-f004:**
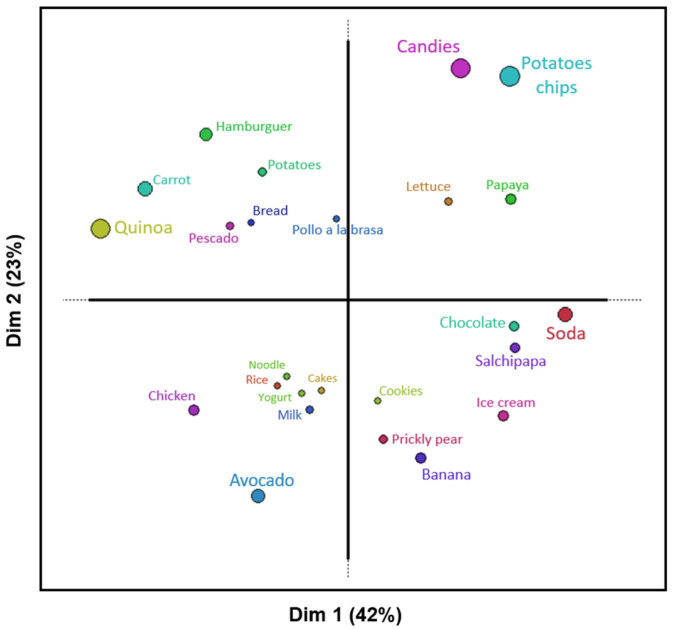
Two-dimensional representation of the foods grouped by the children in Cluster 2 (own elaboration).

**Figure 5 foods-14-00348-f005:**
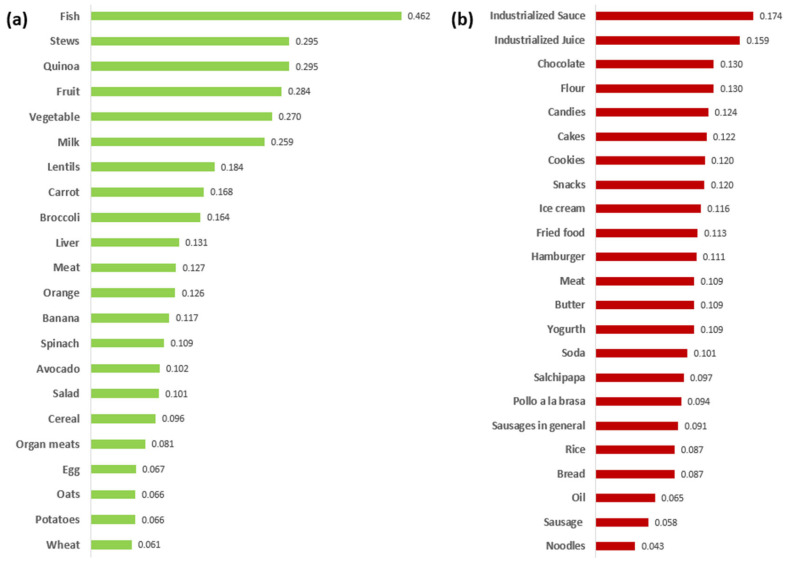
Cognitive salience index of Moquegua parents (**a**) healthy food and (**b**) unhealthy foods (own elaboration).

**Figure 6 foods-14-00348-f006:**
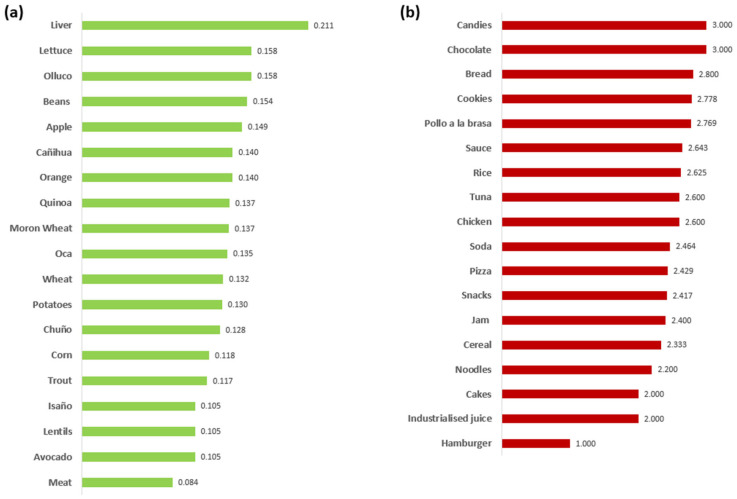
Cognitive salience index of Juliaca–Puno parents for (**a**) healthy foods and (**b**) unhealthy foods (own elaboration).

**Table 1 foods-14-00348-t001:** Distribution of the healthiness of the food consumed in Juliaca and Moquegua, based on Monteiro [32] and MINSA [31].

Healthy	Fairly Healthy	Unhealthy
Milk	Chicken	Ice cream
Fish	Potato	Potato chips
Carrot	Rice	Cake
Lettuce	Bread	Salchipapas (Peruvian dish consists of french fries and fried sausages)
Banana	Noodles	Hamburger
Papaya	Yogurt	Soda
Prickly pear		Pollo a la brasa (Peruvian baked chicken)
Avocado		Biscuits
Quinoa		Candies
		Chocolate

**Table 2 foods-14-00348-t002:** Socio-demographic characteristics of participating parents (own elaboration based on survey data).

Variable	Moquegua Parents	Juliaca Parents
Number of parents (*n*)	60	40
Average age (years)	38.5 ± 6.36	40 ± 7.07
Educational level (a) Completed elementary school (b) Completed high school (c) Incomplete university (d) University complete (e) Postgraduate	3% 7% 18% 64% 8%	10% 28% 38% 22% 2%
Average monthly income (USD)	700 ± 424.26	475 ± 194.45
Average number of children	2 ± 1.41	4.5 ± 1.06

## Data Availability

The data supporting the results of this study are included in the present article and Appendix A. Further inquiries can be directed to the corresponding author.

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
