# Peer review of "Reasons Behind (Un)Healthy Eating Among School-Age Children in Southern Peru"

_foods, 2025, doi:10.3390/foods14030348_

Round 1
Reviewer 1 Report
Comments and Suggestions for Authors
The article seems interesting, but it contains numerous shortcomings.
Substantive comments:
1. What were the reasons for the children's choices?
2. How do parents influence the children's choices?
3. The authors should prepare a table with information about the parents (age, economic situation, education, number of siblings)
4. The title of the work does not reflect the content.
5. The purpose of the work is not properly highlighted.
6. There is no limitation of the study part.
7. Conclusions should be presented in the main points, answering the research questions.
Technical comments:
1. Figures are not properly described, it is necessary to provide a legend for each figure.
2. The bibliography is prepared incorrectly, because each item begins with the first letter of the first name, not the last name. In some items there is no link.
Author Response
Thank you very much for taking the time to review this manuscript. Below you will find the detailed responses and corresponding revisions/corrections in another color (blue) in the resubmitted files.
Substantive comments:
Comments 1: What were the reasons for the children's choices?
Response 1: Thank you for your comment. We agree with your point and have added additional information. The changes can be found on page 6, lines 226-236; page 7, lines 263-267; and page 7-8, lines 280-285.
Comments 2: How do parents influence the children's choices?
Response 2: Thank you for pointing this out. We agree with this comment, so we added additional information. This change can be found on page 11, lines 409-418, 422-432, and on page 12, lines 438-442.
Comments 3: The authors should prepare a table with information about the parents (age, economic situation, education, number of siblings)
Response 3: Thank you for bringing this to our attention. We agree with your comment and have added further information. This change can be found on page 4, lines 131-134, and on page 11, lines 402-406.
Comments 4: The title of the work does not reflect the content.
Response 4: Thank you for your feedback. We appreciate this comment and have added additional information. This change is on page 1, lines 2.
Comments 5: The purpose of the work is not properly highlighted.
Response 5: Thank you for pointing this out. We agree with this comment, so we modified it. This change can be found on page 2, lines 67-70.
Comments 6: There is no limitation of the study part.
Response 6: Thank you for pointing this out. We agree with this comment, so we added additional information. This change can be found on page 12, lines 455-458.
Comments 7: Conclusions should be presented in the main points, answering the research questions.
Response 7: Thank you for pointing this out. We agree with this comment, so we added additional information. This change can be found on page 12, lines 465-479.
Technical comments:
Comments 8: Figures are not properly described; it is necessary to provide a legend for each figure.
Response 8: Thank you for pointing this out. We agree with this comment. Therefore, we modify. This change can be found on page 4, line 120, and page 5, line 188.
Comments 9: The bibliography is prepared incorrectly, because each item begins with the first letter of the first name, not the last name. In some items there is no link.
Response 9: Thank you for pointing this out. We agree with this comment. Therefore, we modify. This change can be found in number 14, line 496
Reviewer 2 Report
Comments and Suggestions for Authors
The article explores the barriers to healthy eating among school-age children in the regions of Moquegua and Juliaca, in southern Peru. The research investigated how children aged 6 to 12 classify foods as healthy or unhealthy using an adapted free sorting task with food images. Simultaneously, parents were interviewed to list healthy and unhealthy foods, enabling the analysis of the Cognitive Salience Index (CSI) for different foods. The results indicated that children from Moquegua demonstrated a greater ability to distinguish healthy foods than those from Juliaca, with clear influence from parental food choices. The study also revealed cultural and geographical differences between the regions, impacting food perceptions. The article concludes that educational interventions can improve nutritional literacy and recommends expanding the study to other regions in Peru. Based on scientific evaluation criteria, I would not approve the article for publication in its current form but recognize its potential for acceptance after significant revisions. Below are the reasons for this decision:
Formatting and Presentation
- Some references lack consistent formatting, e.g., "[3]" appears isolated without proper integration into the text.
- The topic "Free listing of parents" is duplicated (lines 256 and 281), indicating organizational issues.
- Sentences like "In 2023, 37 million children (<5 years old) were overweight [5]" lack smooth integration with the main idea.
- Many sentences are lengthy and could be simplified to improve readability. For instance, the sentence starting with "According to parents' perceptions..." in line 330 is too long and could be split.
- Phrases like "It is possible to affirm..." are literal translations from other languages and sound inappropriate in technical English.
Methodology
- The description of the sorting task lacks clarity on how symbols ("angel," "devil," "happy face") were validated for children.
- The food selection was based on self-reported data from only 30 participants, which is insufficient to represent diverse populations.
- There is no robust justification for using only four categories in the sorting cards, especially when the foods varied greatly in nutritional attributes.
Ethical Consent: Although parents signed consent forms, there is insufficient detail on how anonymity and children’s confidentiality were ensured.
Data and Analysis
- Figures and graphs are inadequately described in the text. It is unclear what "Cluster 1" and "Cluster 2" visually represent, making the results hard to interpret.
- There are no details on statistical tests used to validate the cluster analysis results. Simply applying the Distatis technique is insufficient without describing reliability metrics.
- The formula for calculating the Cognitive Salience Index (CSI) lacks clear examples, hindering replicability.
References
- Many references are cited only by numbers ([1], [3], [4]) without appropriate contextual detail.
- For example, Monteiro et al. (2018) is used to support the NOVA classification (line 202) but could be complemented with more recent studies.
Discussion
- The discussion on parental influence over children’s eating habits is presented as universal, without adequately recognizing regional or socioeconomic variations.
- The article assumes cultural differences between Moquegua and Juliaca explain the discrepancies in food categorization by children but does not explore other factors, such as education or food exposure.
Conclusions
- The conclusions reiterate well-established knowledge (e.g., the importance of healthy food environments) but fail to provide new insights based on the results.
- The recommendation for "studies in other regions of Peru" is generic and lacks specificity regarding methods or variables of interest.
Recommendation
While the article addresses a relevant topic and employs an interesting methodology, its weaknesses in writing, methodology, data analysis, and discussion limit its scientific rigor and contribution. Significant revisions are necessary to address these issues. With these improvements, the manuscript could become a valuable addition to the field of nutritional and cultural studies.
Author Response
Thank you very much for taking the time to review this manuscript. Below you will find the detailed responses and corresponding revisions/corrections in another color (blue) in the resubmitted files.
Formatting and Presentation
Comments 1: Some references lack a consistent format, for example, “[3]” appears in isolation without proper integration in the text.
Response 1: Thank you for pointing this out. We agree with this comment. Therefore, we have removed reference “[3]” and integrated it into line 32.
Comments 2: The item “Free parent listing” is duplicated (lines 256 and 281), indicating organizational problems.
Response 2: Thank you for pointing this out. We agree with this comment, so we have removed line 281.
Comments 3: Sentences such as “In 2023, 37 million children (<5 years) were overweight [5]” lack smooth integration with the main idea.
Response 3: Thank you for pointing this out. We agree with this comment, so we have removed it. This change can be found on page 1, line 32.
Comments 4: Many sentences are lengthy and could be simplified to improve readability. For instance, the sentence starting with "According to parents' perceptions..." in line 330 is too long and could be split.
Response 4: Thank you for pointing this out. We agree with this comment. The text has been revised to improve readability by breaking longer sentences into shorter sentences. This change can be found on pages 9 and 10, lines 349-361.
Comments 5: Phrases like "It is possible to affirm..." are literal translations from other languages and sound inappropriate in technical English
Response 5: Thank you for pointing this out. We agree with this comment. Therefore, we have removed phrases like "It is possible to affirm...". This change can be found on page 7, lines 263-267.
Methodology
Comments 6: The description of the sorting task lacks clarity on how symbols ("angel," "devil," "happy face") were validated for children.
Response 6: Thank you for pointing this out. We agree with this comment, so we added additional information. This change can be found on page 3, lines 107-112.
Comments 7: The food selection was based on self-reported data from only 30 participants, which is insufficient to represent diverse populations
Response 7: Thank you for bringing this to our attention. In this study, we initially examined national food consumption and sales statistics to ensure that the selected foods were representative of the local environment. We then validated our choices with a group of untrained consumers (n=30). Our approach to food selection aligns with similar studies. For example:
Ong and Delarue (2024) utilized a consumer-led perceptual mapping approach with a sample size of 34 participants, divided into two stages. In the first stage, informed consumers pre-selected products. In the second stage, untrained consumers evaluated and discussed their perceptions in focus groups. This design enabled iterative refinement of food selections and ensured that the products remained relevant to the final consumers. (Refer to https://www.sciencedirect.com/science/article/pii/S095032932300263X?via%3Dihub)
Ford et al. (2023) utilized detailed screening questionnaires to categorize participants based on their eating habits, consumption frequency, and sociodemographic information. They organized these participants into small, balanced groups that engaged in structured discussions. This approach illustrates that even with small sample sizes, it is possible to obtain representative data when participants are carefully segmented. (Refer to https://www.sciencedirect.com/science/article/pii/S019566632302487X?via%3Dihub#bib30)
Collier et al. (2021) employed a qualitative method that included pre-tasks, such as Napping®, and semi-structured focus groups. In this approach, participants interacted with related products (without tasting them) to foster discussion about their perceptions and attitudes toward food. This methodology highlights the significance of combining hands-on activities with structured discussions to enhance the data collected, even when working with small, segmented samples. (Refer to https://www.sciencedirect.com/science/article/pii/S019566632100550X?via%3Dihub#bib32)
We are adopting a similar approach. Additionally, we are preparing a larger-scale study that will include a broader population and cover various regions. This next phase will enable us to validate and generalize our current findings, thereby increasing the study's representativeness.
Comments 8: There is no robust justification for using only four categories in the sorting cards, especially when the foods varied greatly in nutritional attributes.
Response 8: Thank you for bringing this to our attention. We based our approach on the preliminary studies by Alfaro et al. (2020) and Varela and Salvador (2014), which utilized the “free sorting task” technique. In our study, we adapted this methodology by presenting the categories as four distinct boxes instead of dividing them on an A3 paper. Additionally, since our participants are school-aged children, introducing more than four options could have posed challenges. Having a larger number of categories might confuse the participants or increase the time required to complete the task, making it more analytical rather than intuitive—an important consideration when working with this age group.
Comments 9: Although parents signed consent forms, there is insufficient detail on how anonymity and children’s confidentiality were ensured.
Response 9: Thank you for pointing this out. We agree with this comment, so we added additional information. This change can be found on page 2, lines 76-78.
Data and Analysis
Comments 10: Figures and graphs are inadequately described in the text. It is unclear what "Cluster 1" and "Cluster 2" visually represent, making the results hard to interpret.
Response 10: The descriptions of the figures have been revised to improve clarity. We hope they are now easier to understand.This change can be found on page 5, lines 169-179
Comments 11: There are no details on statistical tests used to validate the cluster analysis results. Simply applying the Distatis technique is insufficient without describing reliability metrics.
Response 11: We understand the concern regarding the validation of cluster analysis results. The procedure involved analyzing the complete matrix using the Distatis technique to obtain a general representation of consumer relationships. Based on this analysis, RV coefficients were calculated to measure the similarity between individual matrices, and Euclidean distances between consumers were derived. A cluster analysis was then performed, represented in a dendrogram, to classify consumers into homogeneous groups. Subsequently, the Distatis technique was reapplied within each cluster to achieve more homogeneous representations and reduce interindividual differences among the participating children, as recommended in the literature (Næs et al., 2018). This approach ensures that the results are robust and reliable.
Reference: Næs, T., Varela, P., & Berget, I. (2018). Individual differences in projective mapping and sorting data. En T. Næs, P. Varela, & I. Berget (Eds.), Individual differences in sensory and consumer science (pp. 57–72). Elsevier. https://doi.org/10.1016/B978-0-08-101000-6.00004-4
Comments 12: The formula for calculating the Cognitive Salience Index (CSI) lacks clear examples, hindering replicability.
Response 12: Thank you for pointing this out. We agree with this comment. Therefore, we added additional information. This change can be found on page 8, lines 294 - 299 and 310-311. Additionally, we included a new table to present the data, and we applied the same procedure to determine the Cognitive Salience Index (CSI).
References
Comments 13: Many references are cited only by numbers ([1], [3], [4]) without appropriate contextual detail.
Response 13: Thank you for pointing this out. We agree with this comment. Therefore, we added additional information. This change can be found on page 1, line 30; page 5, line164; page 6, lines 200, 213, 223, 232, 235; page 9, line 318; page 10, line 393; page 11, lines 412, 419, 429; page 12, line 444, and 445.
Comments 14: For example, Monteiro et al. (2018) is used to support the NOVA classification (line 202) but could be complemented with more recent studies.
Response 14: Thank you for pointing this out. We agree with this comment, so we added additional information. This change can be found on page 2, line 84.
Discussion
Comments 15: The discussion on parental influence over children’s eating habits is presented as universal, without adequately recognizing regional or socioeconomic variations.
Response 15: Thank you for pointing this out. We agree with this comment, so we added additional information. This change can be found on pages 10 and 11, lines 396-401, 424-429.
Comments 16: The article assumes cultural differences between Moquegua and Juliaca explain the discrepancies in food categorization by children but does not explore other factors, such as education or food exposure.
Response 16: Thank you for pointing this out. We agree with this comment. Therefore, we added additional information. This change can be found on page 11, lines 409-418 and 422-432; page 12, lines 435-442.
Conclusions
Comments 17: The conclusions reiterate well-established knowledge (e.g., the importance of healthy food environments) but fail to provide new insights based on the results.
Response 17: Thank you for pointing this out. We agree with this comment, so we added additional information. This change can be found on page 12, lines 465-479.
Comments 18: The recommendation for "studies in other regions of Peru" is generic and lacks specificity regarding methods or variables of interest.
Response 18: Thank you for pointing this out. We agree with this comment. Therefore, we added additional information. This change can be found on page number 12, line 476-479.
Recommendation
Comments 19: While the article addresses a relevant topic and employs an interesting methodology, its weaknesses in writing, methodology, data analysis, and discussion limit its scientific rigor and contribution. Significant revisions are necessary to address these issues. With these improvements, the manuscript could become a valuable addition to the field of nutritional and cultural studies.
Response 19: The points mentioned have been addressed, and we have implemented the recommended revisions. We hope the manuscript is now improved and meets the expectations for scientific rigor and contribution. Thank you for your constructive feedback.
Round 2
Reviewer 1 Report
Comments and Suggestions for Authors
Dear Authors,
Thank you for revised version. However I have one remarks. In my previous review I asked you about correction your tables and figures connected with authorship. Every figures and tables should have below information about "own collaboration" or "based on ..".
Author Response
Thank you very much for taking the time to review this manuscript. Below you will find the detailed responses and corresponding revisions/corrections in another color (red) in the resubmitted files.
Comments 1: Thank you for revised version. However I have one remarks. In my previous review I asked you about correction your tables and figures connected with authorship. Every figures and tables should have below information about "own collaboration" or "based on ..".
Response 1: Thank you for pointing this out. We agree with this comment, so we added additional information. This change can be found on page 3, lines 98-99; page 4, line 123; page 5, line 191; page 6, line 264 and line 287; page 9, line 362; page 10, line 406 and page 11, line 438.
Reviewer 2 Report
Comments and Suggestions for Authors
- Ensure that terms such as "Cognitive Salience Index" and "Free Sorting Task" are used consistently throughout the text. Occasionally, terms may vary, which can confuse the reader.
- The figures mentioned in the text (e.g., Figures 3 and 4) are described, but a more detailed explanation in the body of the text regarding the visual elements presented in each figure (e.g., the meaning of each color or grouping) would be helpful for less experienced readers.
- The newly added references enrich the text; however, in some sections, they appear abruptly. Try integrating them more naturally to strengthen the arguments presented.
Discussion and Conclusions Sections
While the feedback has led to the addition of more information, the conclusions section could still emphasize the practical implications of the results more robustly, such as providing specific recommendations for public policies or educational programs.
Although some sentences have been revised, there are still lengthy and dense passages, particularly in sections like "Methodology." Break these passages into shorter, more direct sentences to improve the text's readability.
Typographical or Grammatical Errors
Some sentences lack articles or prepositions. For example: "Children classified foods such as fruit, rice, and milk as ‘healthy’ if they consumed them regularly at home or school." This could be revised to: "Children classified foods, such as fruits, rice, and milk, as ‘healthy’ when consumed regularly at home or school."
1. Methodology
- In the explanation of the "Cognitive Salience Index," a practical and numerical example could help readers better understand how it is calculated and applied.
2. Results
- The connection between the cluster results and the demographic/social characteristics mentioned in Table 3 could be clearer. Consider elaborating on how these variables influenced the classification of foods.
3. Discussion
- Beyond cultural and socioeconomic differences, explicitly mention the influence of external factors, such as advertising and the availability of ultra-processed foods, on the formation of food preferences.
4. Conclusions
- Expand on how the results can be applied in public health strategies. Consider including recommendations for school-based interventions or parental education programs.
Author Response
Thank you very much for taking the time to review this manuscript. Below you will find the detailed responses and corresponding revisions/corrections in another color (red) in the resubmitted files.
Comments 1: Ensure that terms such as "Cognitive Salience Index" and "Free Sorting Task" are used consistently throughout the text. Occasionally, terms may vary, which can confuse the reader
Response 1: Thank you for pointing this out. We agree with this comment and have reviewed the manuscript to ensure consistent use of terms. This change can be found on page 2, line 49, 60; pag3, line 100 and 119; pag 4 line 121, 131; pag 5 line 163 and 166; pag 8 line 298, 299, 310, 313; pag. 9 line 365, 367, 370.
Comments 2: The figures mentioned in the text (e.g., Figures 3 and 4) are described, but a more detailed explanation in the body of the text regarding the visual elements presented in each figure (e.g., the meaning of each color or grouping) would be helpful for less experienced readers.
Response 2: Thank you for pointing this out. We agree with this comment, so we added additional information. This change can be found on page 6, lines 195-199 and page 7, lines 263 and 266.
Comments 3: The newly added references enrich the text; however, in some sections, they appear abruptly. Try integrating them more naturally to strengthen the arguments presented.
Response 3: Thank you for pointing this out. We agree with this comment, so we added additional information. This change can be found on page 11 and 12, lines 439, 449 and 460, and so we have removed it. This change can be found on page 6, line 243.
Discussion and Conclusions Sections
Comments 4: While the feedback has led to the addition of more information, the conclusions section could still emphasize the practical implications of the results more robustly, such as providing specific recommendations for public policies or educational programs.
Response 4: Thank you for pointing this out. We agree with this comment so we added additional information. This change can be found on page 12, lines 507-523.
Comments 5: Although some sentences have been revised, there are still lengthy and dense passages, particularly in sections like "Methodology." Break these passages into shorter, more direct sentences to improve the text's readability
Response 5: Thank you for pointing this out. We agree with this comment. This change can be found on page 2, lines 72 -159.
Typographical or Grammatical Errors
Comments 6: Some sentences lack articles or prepositions. For example: "Children classified foods such as fruit, rice, and milk as ‘healthy’ if they consumed them regularly at home or school." This could be revised to: "Children classified foods, such as fruits, rice, and milk, as ‘healthy’ when consumed regularly at home or school."
Response 6: Thank you for pointing this out. We agree with this comment and have revised the sentence. This change can be found on page 6, lines 232-233
Methodology
Comments 7: In the explanation of the "Cognitive Salience Index," a practical and numerical example could help readers better understand how it is calculated and applied.
Response 7: Thank you for pointing this out. We agree with this comment so we added additional information. This change can be found on page 8, lines 300-307.
Results
Comments 8: The connection between the cluster results and the demographic/social characteristics mentioned in Table 3 could be clearer. Consider elaborating on how these variables influenced the classification of foods.
Response 8: Thank you for pointing this out. We agree with this comment so we added additional information. This change can be found on page 11 and 12, lines 429-431, 443-447, 463-476
Discussion
Comments 9: Beyond cultural and socioeconomic differences, explicitly mention the influence of external factors, such as advertising and the availability of ultra-processed foods, on the formation of food preferences.
Response 9: Thank you for pointing this out. We agree with this comment, so we added additional information. This change can be found on page 12, lines 492-498.
Conclusions
Comments 10: Expand on how the results can be applied in public health strategies. Consider including recommendations for school-based interventions or parental education programs.
Response 10: Thank you for pointing this out. We agree with this comment, so we added additional information. This change can be found on page 12 and 13, lines 518-523.
Round 3
Reviewer 2 Report
Comments and Suggestions for Authors
The authors have made the requested changes.
Thank you.
Author Response
Thank you for your feedback, we appreciate the valuable contributions that have helped to improve the quality of the manuscript.